# Computational and Histological Analyses for Investigating Mechanical Interaction of Thermally Drawn Fiber Implants with Brain Tissue

**DOI:** 10.3390/mi12040394

**Published:** 2021-04-02

**Authors:** Kanghyeon Kim, Changhoon Sung, Jungjoon Lee, Joonhee Won, Woojin Jeon, Seungbeom Seo, Kyungho Yoon, Seongjun Park

**Affiliations:** 1Department of Bio and Brain Engineering, Korea Advanced Institute of Science and Technology (KAIST), Deajeon 34141, Korea; ktotheh0@kaist.ac.kr (K.K.); lluccidd@kaist.ac.kr (C.S.); arsow26@kaist.ac.kr (W.J.); 2Program of Brain and Cognitive Engineering, Korea Advanced Institute of Science and Technology (KAIST), Deajeon 34141, Korea; jungjoon@kaist.ac.kr (J.L.); juneduke@kaist.ac.kr (J.W.); 3Department of Electrical Engineering, Ulsan National Institute of Science and Technology (UNIST), Ulsan 44919, Korea; ssb3304@unist.ac.kr; 4Center for Healthcare Robotics, Korea Institute of Science and Technology (KIST), Seoul 02792, Korea; 5Division of Bio-Medical Science and Technology, KIST School, Korea University of Science and Technology, Seoul 02792, Korea; 6KAIST Institute of Health Science and Technology (KIHST), Korea Advanced Institute of Science and Technology (KAIST), Daejeon 34141, Korea

**Keywords:** fiber neural probes, TDP, FEA, soft materials, IHC

## Abstract

The development of a compliant neural probe is necessary to achieve chronic implantation with minimal signal loss. Although fiber-based neural probes fabricated by the thermal drawing process have been proposed as a solution, their long-term effect on the brain has not been thoroughly investigated. Here, we examined the mechanical interaction of thermally drawn fiber implants with neural tissue through computational and histological analyses. Specifically, finite element analysis and immunohistochemistry were conducted to evaluate the biocompatibility of various fiber implants made with different base materials (steel, silica, polycarbonate, and hydrogel). Moreover, the effects of the coefficient of friction and geometric factors including aspect ratio and the shape of the cross-section on the strain were investigated with the finite element model. As a result, we observed that the fiber implants fabricated with extremely softer material such as hydrogel exhibited significantly lower strain distribution and elicited a reduced immune response. In addition, the implants with higher coefficient of friction (COF) and/or circular cross-sections showed a lower strain distribution and smaller critical volume. This work suggests the materials and design factors that need to be carefully considered to develop future fiber-based neural probes to minimize mechanical invasiveness.

## 1. Introduction

Among many types of neural interfaces, implantable neural probes are in the limelight because of their capability to record neural signals with high resolution [1,2,3]. However, the long-term application of these probes was found to be challenging due to the foreign body response (FBR) and glial scarring that follows the implantation of a probe. These complications are known to increase the interfacial impedance between the electrodes and neural tissue, which, in turn, hinder the electrical recording and stimulation functionalities of the probe [4,5,6]. One of the main causes of FBR is the mechanical mismatch between the implanted probes and brain tissue [7], which causes larger relative displacements during micromotion, aggravating the tissue damage [8,9,10,11]. Accordingly, more interest is being given to developing flexible neural devices based on soft materials [12,13,14,15,16].

Recently, to address such challenges, fiber implants developed by a novel fabrication method, thermal drawing process (TDP), have been proposed [4,10]. TDP grants neural implants some distinct advantages: (i) the material choice is not limited by fabrication conditions, and thus, multiple softer materials can easily be integrated into implants; (ii) implants can have various design factors such as cross-sectional structures and geometric features because these factors of the macroscopic preform are maintained during the process. Due to these advantages, multifunctional fiber implants manufactured with TDP are being applied in a wide scope of fields including optogenetics, electrophysiology, drug delivery, etc. [17,18,19,20].

Despite the recent development of various fiber implants [21,22,23], however, existing studies on probe design only focus on conventional shank-shaped probes fabricated with stiff materials [11,24,25,26,27,28]. Therefore, the mechanical effects of the material and the design factors of fiber implants on neural tissue remain unclear. This emphasizes the need for a comprehensive study on multimaterial and multistructured fiber implants.

In this study, we conducted computational and histological analyses to examine the mechanical effect of the base materials and design factors of fiber implants on brain tissue. We also attempted to provide a comprehensive framework about the materials and the design factors of the fiber implants (Figure 1). A finite element model (FE model) and immunohistochemical analysis (IHC) were employed to provide an integrated overview of fiber implants. Our results suggest critical factors in designing the next-generation fiber-based neural probes.

## 2. Materials and Methods

### 2.1. Preparation of Neural Implants

Fiber implants were fabricated with four different materials: stainless steel (steel), silica, polycarbonate (PC), and HydroMed D4 (hydrogel) (Figure 2). All fiber implants were manufactured to a similar diameter in the range of 400 ± 30 μm. The steel implants were obtained by purchasing microwires from Goodfellow FF215135 (Huntingdon, UK), and the silica implants were obtained by stripping the cladding from commercial silica-core optical fibers (FT400UMT, Thorlabs, Newton, NJ, USA). For the PC implant preparation, after degassing at 80 °C for 2 weeks, 10-mm-thick rods were thermally drawn into a fiber using a custom-built fiber drawing tower at a drawing temperature of 150–160 °C. HydroMed D4, an ether-based polyurethane hydrogel (AdvanSource Biomaterials Corp., Wilmington, MA, USA), was selected for its thermoplasticity, as it ensures the stable manufacturing of the hydrogel fibers. The preform was prepared by solvent casting the Hydromed D4 into a thin sheet and then rolling it into a 10-mm-thick rod. Preforms were then drawn using the same custom tower at drawing temperatures of 80–90 °C. To eliminate the effect of surface properties including the mechanical friction and chemical adhesion, all implants were dip-coated with polydimethylsiloxane (PDMS, Sylard 184, Dow Corning Midland, MI, USA). The base and curing agents of the PDMS were mixed to a 10:1 ratio.

### 2.2. Finite Element Analysis (FEA)

To assess mechanical interactions between implanted neural probes and rodent brain tissue, 3D FEA was performed with appropriate material properties and friction coefficients. All the simulations were carried out using the widely used commercial FEA software ANSYS 2019 R1 (ANSYS Inc., Canonsburg, PA, USA) on a quad-core workstation (Intel^®^ Core™ i5-9600KF CPU @ 3.70 GHz, 32 GB RAM, Microsoft Windows 10 Pro 64 bit).

#### 2.2.1. Geometry and Interface

A cylindrical probe and surrounding brain tissue were modeled as shown in Figure 3A. The implanted probe was modeled as a cylinder with a diameter of 400 μm and a length of 3 mm (*z*-axis). The target brain tissue was modeled as a cuboid with a thickness of 1 mm, a width of 1 mm, and a depth of 6 mm. Additionally, a void space was modeled in the cuboid to fit the cylindrical probe. The length of the probe model was chosen as 3 mm based on the depth of the target region (i.e., the hippocampal region depicted in Figure 3B) of the surgery. The dimensions of the brain tissue model were determined to sufficiently cover the range that reportedly shows dramatic decreases in neuron density (60 μm from the electrode) and recordable range (140 μm) [11,29,30]. The probe and brain tissue were discretized by hexahedrons and tetrahedrons, respectively. A hexahedral-dominant meshing algorithm [31] was used because hexahedrons generally show better performance [32]. Domains of both probe and brain tissue were set to the average mesh size of 80 μm [33]. Around the contact surface between the probe and brain, we applied further refinement of the mesh size as 23.5 μm; this condition was confirmed to have a total equivalent strain error of less than 5% through the mesh convergence test. Each mesh was modeled as quadratic finite elements with displacement–pressure mixed formulations (i.e., SOLID 186 and 187 in ANSYS), which show a superior solution accuracy when analyzing incompressible material (i.e., brain tissue in this study). A total of 744,018 DOFs, including 1985 elements for the probe and 71,930 elements for the brain tissue, were used. More detailed information on the number of used elements, nodes, and DOFs is shown in Table 1.

To represent the mechanical interaction between the implanted probe and brain tissue, a surface-to-surface contact model was utilized. The entire enclosure and bottom area of the cylindrical probe were set as a target surface, and the inner wall of the void space in the brain tissue model was defined as a contact surface. The contact elements, modeled by TARGE170 for the target surface and CONTAC174 for the contact surface, were overlaid on the target and contact surface of the FE model, respectively. The contact interaction was formulated using the augmented Lagrange multiplier method with contact detection at the Gauss integration points. The contact normal stiffness was automatically determined based on the material properties and the size of the underlying elements. The Coulomb friction model with a coefficient of friction (COF) of 0.3, which is in the general range of COF between the sample and the soft tissue including the brain [34,35], was used for modeling tangential forces between the contacting surfaces.

#### 2.2.2. Material Model

The nonlinear stress–strain behavior of the brain tissue was represented using the Ogden hyperelastic material model with assumptions of isotropic, homogeneous, and incompressible material [27,36,37,38,39]. The strain energy density function of the Ogden hyperelastic material model is given as Equation (1) [40,41]
(1)$$W=\overset{N}{\underset{i=1}{\sum }}\frac{\mu_{i}}{\alpha_{i}}\left({\bar{\lambda }}_{1}{}^{\alpha_{i}}+{\bar{\lambda }}_{2}{}^{\alpha_{i}}+{\bar{\lambda }}_{3}{}^{\alpha_{i}}-3\right)\;with\;{\bar{\lambda }}_{l}=J^{-\frac{1}{3}}\lambda_{l}$$
where W is the strain energy potential, *N* is the order of the model, $\mu_{i}$ and $\alpha_{i}$ are material constants, *J* is the determinant of the elastic deformation gradient, and $\lambda_{l}$ is the principal stretches of the left Cauchy–Green tensor. In this study, we used the first-order Ogden model (i.e., *N* = 1) with parameters *μ*_1_ = 8.1 and *α*_1_ = 15.7 to represent our implant target region (CA3SR region of the hippocampus in the adult rodent brain) [41]. To exclude the volumetric response term of the Ogden hyperelastic material model in ANSYS, we applied a penalty value to the corresponding parameter, which is confirmed to be large enough to constrain the volumetric term.

The probe materials were represented by the linearly elastic material model as the large difference in elastic modulus between the probe, and the tissue makes the deformation of the probe small enough to ignore the linearity during micromotions [42]. The material properties of each of the probes are listed in Table 2. Mechanical properties of the conventional materials (steel, silica, and PC) were provided by the corresponding suppliers and references. Specifically, for the hydrogel, Poisson’s ratio was measured using the INSTRON 3367 universal testing machine (Instron Co. Ltd., Norwood, MA, USA) at a speed of 10 mm/min. This measurement was performed with hydrogel films swelled in phosphate-buffered saline (PBS) overnight under the assumption of a fully hydrated condition in the brain. The true density of the hydrogel was measured using the AccuPyc II 1340 pycnometer (Micromeritics Inc., Norcross, GA, USA). The density used in the simulations was then estimated from the true density measured at room temperature and the water content property (50%) given by the supplier.

#### 2.2.3. Boundary Conditions and Solution Scheme

All the degrees of freedom on the top surface of the probe were constrained to represent a probe fixed to the skull. According to the experimental study by Gilletti and Muthuswamy [43], the brain micromotion of an anesthetized rodent was modeled by the superposition of movement induced by vascular pulsatility and respiration. The mean displacements of brain tissue without dura were 2.0 μm related to vascular pulsatility and 11.4 μm related to respiration, showing the highest correlation at the frequency of 4 and 1 Hz, respectively. Therefore, the superimposed sinusoidal displacement of 11.4sin(2πt)+2sin(8πt) μm (where *t* represents the time) was prescribed to each side of the brain tissue model perpendicularly in 3D.

The linear implicit transient analysis was carried out through the Hilber–Hughes–Taylor (HHT) time integration method with the full Newton–Raphson solution procedure [44]. The total physical time of 1 s was simulated because the result for the 1 s was confirmed to be repeated over an extended period. The size of the time step was determined by an automatic time-stepping algorithm with an initial value of 1 ms. The step size was limited to the minimum value of 0.1 ms and the maximum size of 10 ms.

#### 2.2.4. Assessment of Simulation Results

To assess the effect of the mechanical interactions between the implanted probe and brain tissue, equivalent strain (known as von Mises effective strain) was investigated because it has been explained to be relevant to the chronic immune response in biological tissue [11,45,46]. In past in vitro experiments, significant astrocyte degradation was reported in cyclical strain regimens over 5% [47], and this threshold is considered as the critical strain for the neural “kill zone” in a previous simulation study [36]. Accordingly, we also considered the strain of 5% as the threshold of the “critical region” that causes a significant immune response.

In this study, the equivalent strain values were obtained from the region of the *z*-axis of −3 to 2.9 mm in the tissue model, which excludes the top surface due to inconsistent, asymmetric outliers occurring near the fixed boundary condition (Figure 4A). As the tip of the probe is generally the recording site. We defined the region of interest (ROI) in the modeled brain tissue as 500 μm from the tip. We obtained 2 kinds of equivalent strain data: (1) the equivalent strain versus time and (2) the maximum equivalent strain versus position. For the former, the maximum and average equivalent strain values versus time were obtained from the nodes in the ROI. Specifically, the change of the peak value was observed and compared for each condition from the plot of equivalent strain versus time. For the latter, the maximum values of equivalent strain during the entire simulated period were obtained for each node. Here, we estimated the size of the critical region with the volume of the mesh components that experienced “critical strain” (defined as over 0.05) at least once. This threshold was considered as the mark of irreparable, critical damage inflicted on the region’s neurons. For the calculation of the volume, we constructed an alpha shape (α-shape) [48,49]. The radius of the α-shape was decided by comparing the smallest radius enclosing all of the points that were marked as critical and the maximum characteristic length [31] in the given mesh then choosing the smaller of the two. In addition, the effect of the flexibility of the material was investigated with three paths: top (Z = 2.9 mm), mid (Z = 1.5 mm), and tip (Z = 0 mm) across the cross-section of an x–y plane along the diagonal direction indicated by gray lines in Figure 4A.

The mechanical effect depending on the cross-sectional geometric design of the neural probe was also investigated with 2 kinds of factors: (1) aspect ratio and (2) shape. We designed 6 different probes with the cross-sectional area and height fixed, and the average mesh size was also controlled to be the same (Figure 4B). For the aspect ratio, the equivalent strain values in the previously defined ROI were obtained and compared with the ellipses with aspect ratios of 1:1/1:2/1:3. To investigate the effect of the shape, circle, square, octagon, and rounded square were compared in the same way. The octagon consisted of 5 squares and 4 right-angle isometric triangles, and the rounded square consisted of 5 squares and 4 quadrants.

### 2.3. Implantation Procedure

All animal procedures followed the guidelines and regulations for rodent experiments provided by the Institutional Animal Care and Use Committee (IACUC) of KAIST and were approved by the same committee. Eighteen male C57BL/6NHsd mice aged 6–8 weeks (KOATECH, Gyunggi-do, South Korea) underwent bilateral implantation surgery under aseptic conditions. All surgical equipment was sterilized using steam and 70% ethanol. The probe fibers were also sterilized using 70% ethanol and rinsed with sterile 1× PBS prior to the implantation. The probes for the implantation were pseudorandomly distributed to the mice so that the same number of samples exist for each of the materials (steel, silica, PC, and hydrogel) and were examined after 3 days (*n* = 4) or 4 weeks (*n* = 5) after the implantation. The anesthesia was induced with 4% isoflurane in 100% oxygen (flow rate: 0.8–1.0 L/min) until the righting reflex was no longer observable and maintained throughout the surgery with 1.5–2% isoflurane. Following the anesthesia induction, all following steps of the implantation surgery were performed on a stereotaxic frame (Digital Stereotaxic Instrument 68025, RWD Life Science Corp., Shenzhen, China). After a skin incision to expose the skull, bilateral hippocampal regions were identified on the skull using lambda and bregma points (coordinates relative to the bregma: −2.8 mm anteroposterior (AP); ±3 mm mediolateral (ML); −3 mm dorsoventral (DV), Figure 3B), and a hole was created with a dental drill (1RF HP REF 310 104 001 001 007, Meisinger, Düsseldorf, Germany). The dura mater was removed with fine tweezers, and the area was washed with sterile saline. The probes were lowered manually into the target region via ventral direction. All fiber probes were rigid enough to penetrate the cortex and no additional implantation techniques were used. Finally, the implanted probes were fixed to the skull with an adhesive (Super-Bond C & B, Sun Medical, Shiga, Japan) and dental acrylic cement (Ortho-Jet, Lang Dental, Wheeling, IL, USA).

### 2.4. Immunohistochemistry Procedure

To assess the FBR, immunohistochemical analysis was conducted with anti-GFAP (astrocytes), anti-CD68 (activated microglia and macrophages), anti-Iba1 (all microglia and macrophages), and anti-IgG (BBB breach) antibodies. Goat anti-GFAP 1:1000 (Abcam, ab53554), goat anti-Iba1 1:500 (Abcam, ab107159), rabbit anti-CD68 (ED1) 1:250 (Abcam, ab125212), and donkey anti-mouse-IgG conjugated to Alexa Fluor 568 1:1000 (Thermo Fisher Scientific, Waltham, MA, USA, A10037) primary antibodies were used. For secondary antibodies, donkey antigoat with Alexa Fluor 488 1:500 (Thermo Fisher Scientific, A11055), donkey antigoat with Alexa Fluor 488 1:1000, and donkey antirabbit with Alexa Fluor 594 1:1000 (Thermo Fisher Scientific, A21207) were used. The combination of antibodies is summarized in Table 3.

Mice were anesthetized via an IP injection of ketamine (100 mg/kg) and xylazine (10 mg/kg) in 1× PBS and transcardially perfused with 4% paraformaldehyde (PFA) in phosphate-buffered solution (PBS). The brains were extracted and stored in 4% PFA solution overnight at 4 °C and in 30% sucrose in PBS. The brains were then frozen in an OCT compound (Tissue-Tek 4583; Sakura Finetek Inc, Torrance, CA, USA) and sliced into 40 μm coronal sections using a cryostat (Leica Microsystems, Wetzlar, Germany). The sections were washed three times in 1× PBS each for 10 min and permeabilized and blocked with a 5% normal donkey serum (NDS) and 0.3% Triton X-100 in PBS for 2 h at room temperature. The sections were then washed again three times in 1× PBS each for 10 min and incubated in the primary antibodies in a 5% NDS in PBS solution at 4 °C for 16 h. Following the primary incubation, the sections were washed four times in 1× PBS each for 10 min and incubated in the secondary antibodies in a 5% NDS in PBS for 2 h at room temperature. Following the secondary incubation, the sections were washed four times in 1× PBS each for 10 min and mounted onto microscopic slides with the mounting medium with DAPI, 4′,6-diamidino-2-phenylindole (Vectashield, Vector Laboratories, Burlingame, CA, USA).

A laser-scanning confocal microscope (C2; Nikon, Tokyo, Japan) with 10× (air, NA = 0.45) objective was used for image acquisition and mosaic images were obtained with NIS elements AR software (Nikon). The images of serial z-stack within a depth of 5.4 μm were combined through maximum projection and converted to 16-bit tagged image files (TIFs). ImageJ and a custom MATLAB algorithm (MathWorks) were used to quantify and analyze the fluorescent intensity profiles of IgG, GFAP, Iba1, and CD68. For image analysis, 10 regions of interests (ROIs) in a rectangular shape with a height of 50 μm and a width of 2348 μm (ROI: 500 μm × 2348 μm) were drawn and stacked upward from the center of the implant tip, which was manually defined. Regions of explanted probes were identified by the difference of fluorescence intensity from each ROI. The intensity values in each ROI were averaged along the height to create an intensity profile along the horizontal direction. Then, the intensity profiles from both sides of the implant site were averaged according to the distance from the probe to draw a distance-averaged intensity plot. In each ROI, the probe region was detected, and the average intensity plot against the distance from the probe surface was obtained. Finally, all 10 ROI profiles were averaged to produce a total average intensity plot for the probe.

### 2.5. Statistical Analysis

In each ROI, the distance-intensity profile was normalized to the background defined by the regions more than 500 μm away from the probe surface. The ROIs were then binned into ten 50 μm intervals from the probe surface, and the intensity of each bin was averaged. For these binned intensity plots, a two-way ANOVA and post-hoc Tukey’s test were performed with the averaged intensity between the probe material and the distance from the probe surface.

The mean intensity value in a wide region that includes most of the implant site was obtained from each sample. The region was drawn in a rectangular shape with 750 μm height (550 μm above and 200 μm below from the tip) and 1000 μm width, aligning the center of the probe at the horizontal center of the image. A one-way ANOVA and post-hoc Tukey’s test were performed with these mean values.

## 3. Results

### 3.1. Effect of the Base Materials

The simulation results for the four different material conditions were compared. Figure 5 displays the maximum and averaged equivalent strain versus time in the ROI. In the comparisons of the maximum strain (Figure 5A), the steel, silica and PC probe showed nearly identical results (peak value: 0.1653), whereas the hydrogel (peak value: 0.1156) yielded comparably lower strain levels. The averaged equivalent strain in the ROI gave similar results in that only the hydrogel showed a noticeable difference in value as follows: 0.0161 (hydrogel) < 0.0232 (PC) < 0.0233 (steel and silica) (Figure 5B).

The distributions of the maximum equivalent strain values versus position were compared for each probe material. Figure 6A–D shows that the critical region where the strain is larger than 0.05 (denoted by red in the figures) was intensively distributed in the top surface and the tip of the implant. In particular, the volume of the critical region in the ROI near the tip was notably lower for hydrogel (0.0015 mm^3^) than others (0.0197 mm^3^) (Figure 6H). For three diagonal paths shown in Figure 4A (top (at z = 2.9 mm), mid (at z = 1.5 mm), and tip (at z = 0 mm)), the peak equivalent strain versus position was plotted against the distance from the implanted probe (Figure 6E–G). To investigate the strain distribution by the distance from the probe, the values of both sides of the implant were averaged. The results show that the strain profiles were universally identical at the top path. However, in the other paths (i.e., mid and tip), we were able to observe a difference that is significant only in the hydrogel implant: (1) peak value at the mid path: steel and silica (0.0527) > PC (0.0527) > hydrogel (0.0462); (2) peak value at the tip path: steel, silica and PC (0.1107) > hydrogel (0.0766).

### 3.2. Effect of the Friction Coefficient

The simulation results varying the COF between the neural probe and tissue were analyzed. Figure 7 displays the maximum equivalent strain versus time and the volume of the critical region in the ROI under COF conditions of 0.1, 0.3, 0.5, 0.7, and 0.9. The maxima of the equivalent strain plots showed that a lower COF induced a higher peak strain value as follows: peak values for COF of 0.1, 0.3, 0.5, 0.7, and 0.9 were 0.13176, 0.1157, 0.1070, 0.1025, and 0.0979, respectively. Furthermore, a lower COF induced a wider critical region as follows: volume of the critical region for COF of 0.1, 0.3, 0.5, 0.7, and 0.9 were 0.0026, 0.0015, 0.0011, 0.0008, and 0.0007 mm^3^.

### 3.3. Effect of the Geometry

The effect of geometric factors on the probe was investigated in the same way as above. Firstly, the effect of the aspect ratio was investigated with an elliptical cross-section (Figure 8A,B). The peak value of the maximum strain was increased with higher aspect ratios: peak value: 0.1157 (1:1), 0.1307 (1:2), and 0.1660 (1:3). Furthermore, higher aspect ratios caused a larger volume of the critical region, as follows: 0.0079 mm^3^ (1:3), 0.0048 mm^3^ (1:3), and 0.0015 mm^3^ (1:1). The effect of shape variations was also investigated, using a fixed aspect ratio of 1:1 (Figure 8C,D). The peak value of the maximum strain decreased in the order of octagon, square, rounded square, and ellipse: 0.1385, 0.1236, 0.1226, and 0.1157, respectively. In addition, the round shapes (circle: 0.0015 mm^3^; rounded square: 0.0014 mm^3^) resulted in smaller critical regions than the sharper shapes with edges (square: 0.0019 mm^3^; octagon: 0.0024 mm^3^).

### 3.4. Histological Analysis

In order to elucidate the effect of various elastic moduli of fiber implants on the FBR and validate the reliability of our computational analysis, we conducted histological analyses after the implantation of fiber implants. For the four types of implants (steel, silica, PC, and hydrogel), the acute (three days) and chronic (four weeks) phases of FBR were evaluated with four immunomarkers associated as the following: BBB breach (immunoglobulin G (IgG)); astrocytes, which make up the glial scar (glial fibrillary acidic protein (GFAP)), and microglia/macrophages; which play a key role in the central nervous system’s immune response [50] (the ionized calcium-binding adapter molecule (Iba1) for all microglia/macrophages, including resting cells and macrosialin (CD68) for activated microglia/macrophages) (Figure 9).

We obtained and evaluated two kinds of data: mean value of intensity and intensity–distance profile. The results and analysis of the ANOVA and post-hoc Tukey’s tests are given here. At four weeks after implantation, steel elicits significantly higher overall activation of microglia/macrophages compared to the other materials (*p* < 0.001 in CD68 compared to silica, PC, and hydrogel and *p* < 0.05 in Iba 1 compared to PC and hydrogel) (Figure 10A–D). Specifically, at the distance of 0–100 μm, where it is known that the microglia line up in a densely packed layer [51,52], steel also shows significantly higher immunoreactivity of CD68 (*p* < 0.05) and Iba1 (*p* < 0.05) compared to the others (Figure 10E–H). Moreover, hydrogel induced significantly less glial scarring than steel and PC and less microglial activation than PC in the range of 0–50 μm (*p* < 0.05 for both GFAP and CD68).

At three days post implantation (Figure 11), the immunostaining results showed no significant trend in relation to the materials. This suggests that there was no strong evidence for the correlation between the acute foreign body response and the elastic modulus of base materials, which is different from the results of the four weeks post implantation.

The computational and immunological results were compared to assess the correspondence between the analyses. We averaged the values of the maximum equivalent strain versus position according to the distance in the ROI and overlaid the minimum–maximum-normalized equivalent strain and intensity graphs in Figure 12. We were able to confirm that for both the immunological and simulated calculations, the magnitude of response had a consistent descending order: (steel–silica–PC–hydrogel). The results of GFAP in particular, which is one of the main markers in a chronic immune response because it signals the construction of glial scars, showed very close tendencies in terms of both the magnitude and peak values with the computational result.

## 4. Discussion

Fiber-based neural implants have been receiving increasing attention because of their tendency to induce lower brain tissue damage. Furthermore, their novel manufacturing schemes allow a wide range of material and design choices. However, there have been few studies conducted that investigate the mechanical effects of their material and design factors on brain tissue. Therefore, in this study, we examined the mechanical effect of fiber implants on brain tissue through computational and histological analyses to build and provide a comprehensive framework for designing them.

First, the effect of base materials was investigated with fiber implants fabricated with four different materials: steel, silica, PC, and hydrogel. In the computational analysis, we focused on the peak value of equivalent strain (how high the strain was) and the size of the critical region where the induced strain was over 0.05 (how widespread the high strain values are). Lower strain and smaller critical regions were observed from hydrogel fiber implants compared to all other cases (Figure 5 and Figure 6). Furthermore, in the IHC analysis, hydrogel probes also showed a significantly decreased chronic FBR (Figure 10). These results suggest that in order to decrease the chronic response, extremely soft materials with elastic moduli in the few MPa range are preferable to conventionally used polymers, which tend to have moduli in the ~GPa scale. Nevertheless, the use of hydrogel has been limited to coating materials until recently [53,54,55] due to the absence of a suitable microfabrication technique. This hurdle can be overcome by the utilization of TDP.

Such advantages of soft materials were not so obviously seen in the acute FBR analyses. This is likely due to different factors playing a dominant role in each phase of the implantation, namely the insertion damage for the acute response and the micromotion for the chronic response. More specifically, in the actual insertion process, most of the damage and the concurrent immune responses can be expected to result from the vertical wound that is caused by the probe moving into the tissue. On the other hand, the chronic response occurs after the initial steps of the damage and is hence more related to the micromotions of the probe, which is already embedded in the tissue. Therefore, the material’s bending stiffness displays more impact in the chronic FBR. This assumption can be further verified by conducting additional experiments. For example, one can investigate the postimplantation IHC at more time points to draw a specific picture of the dominant factors in each step of the implantation process. A more direct approach would be to actually specify the effect of each factor in the FBR via varying each of the parameters to find a decoupled trend.

The effects of the design factors of the fiber implants were also investigated by varying the COF and the geometric factors of the cross-section (aspect ratio and shape). Increasing the COF lowered the peak value of equivalent strain and reduced the critical region (Figure 7). This implies that a higher adhesion between the probe and the tissue decreases potential mechanical damage. This dominant influence of friction on the critical region was also stressed in the studies of the importance of coating materials [25,56,57]. We also observed that a relatively symmetrical and round cross-section design induced less strain and produced a smaller critical region by inducing less mechanical damage (Figure 8). This is consistent with a previous study that cylindrical shapes produce significantly diminished inflammatory responses when compared to planar electrodes [58].

Figure 13 shows a graph of the comparison between the result from computational and histological analyses. The maximum-normalized 3D plot of the logarithm of the modulus, peak values of the equivalent strain calculated from computational analysis, and fluorescence intensity of each immune marker for four different materials (steel, silica, PC, and hydrogel) are given. With the comparison plots by distance (Figure 12), this graph shows that the computational and immunological results follow a similar trend, which supports that the chronic immune response is highly related to the mechanical response including deformation of brain tissue and deflection of the probe due to the brain micromotion. This demonstrates the effectiveness of applying FE models in designing neural probes, especially in that one can assess and compare the chronic damage before actual implantations.

For FEA, previous studies generally used one-dimensional (usually longitudinal or transverse) or static load and inverse loading situations [25,26,36,56,59], which applies a load on the top surface of the probe and fixes the bottom surface of brain tissue for efficient calculation. In this study, a more realistic finite element model through a 3D superposed dynamic micromotion input and inverse loading situation was utilized with the addition of biological validation for the reliability of the model. With the new model, we validated that the superposed sine waves do indeed have a significant impact on the strain plot, showing that previous simple load designs are not sufficient for realistic analysis. In addition, we observed a high concentration of strain around the top surface for all cases, whereas previous studies with inverse symmetry show only concentration at the tip region. This difference is due to the fixed boundary condition in our inverted analysis, which is closer to actual in vivo cases, as neural probes are more likely to be fixed to the skull. Therefore, we strongly believe that our improvements in the FE model are critical to generating more reasonable, realistic insights into the mechanical interaction between neural probes and brain tissue.

In summary, we investigated the materials and design factors of fiber-based neural probes to minimize the mechanical effect on the brain tissue. The study revealed that the use of extremely flexible materials with low elastic modulus such as hydrogel would significantly reduce strains in the brain, which is related to lower chronic FBR. Furthermore, an adhesive coating to increase COF in the surface of the probes and a cross-section geometry with a lower aspect ratio and circular shape would also be beneficial in reducing chronic damage. We propose these results along with the holistic analytic methods used as a guideline for developing future chronic neural probes.

## Figures and Tables

**Figure 1 micromachines-12-00394-f001:**
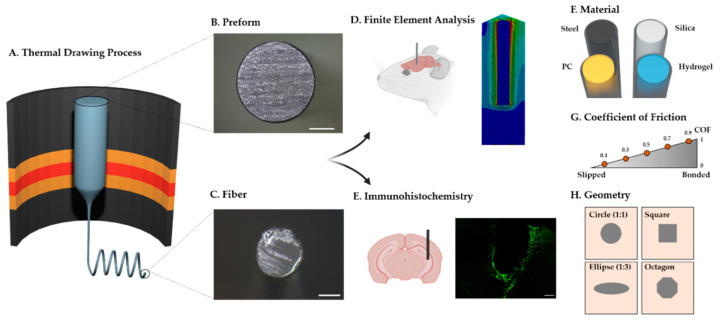
Schematic representation of the study. (**A**) Illustration of the fiber fabrication with the thermal drawing process (TDP). (**B**,**C**) Representative cross-sectional image of the preform (**B**) and the thermally drawn fiber (**C**) made of polycarbonate (the scale bar indicates 3 cm (**B**) and 200 μm (**C**)). (**D**) Finite element analysis and (**E**) histological analysis (immunohistochemistry) after the implantation of fiber implants. (**F**) Illustration of PDMS dip-coated fiber implants with four different materials; stainless steel (steel), silica, polycarbonate (PC), and HydroMed D4 (hydrogel). (**G**,**H**) Illustrations representing design factors considered in this study: coefficient of friction and geometry of the cross-section. In (**H**), the outer square represents the cross-section of the brain tissue, and the inner shape depicts the cross-section of the neural probe.

**Figure 2 micromachines-12-00394-f002:**
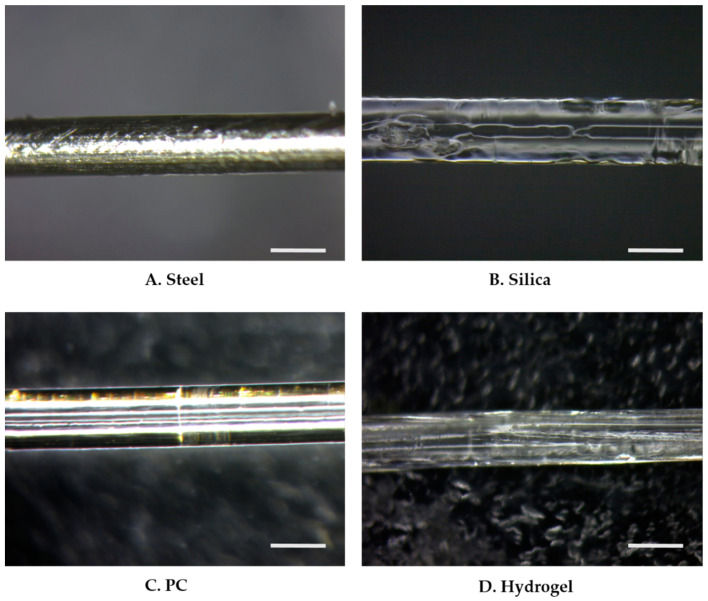
Microscope images of PDMS dip-coated thermally drawn fibers of (**A**) stainless steel (steel), (**B**) silica, (**C**) PC, and (**D**) HydroMed D4 (hydrogel). The scale bar indicates 400 μm.

**Figure 3 micromachines-12-00394-f003:**
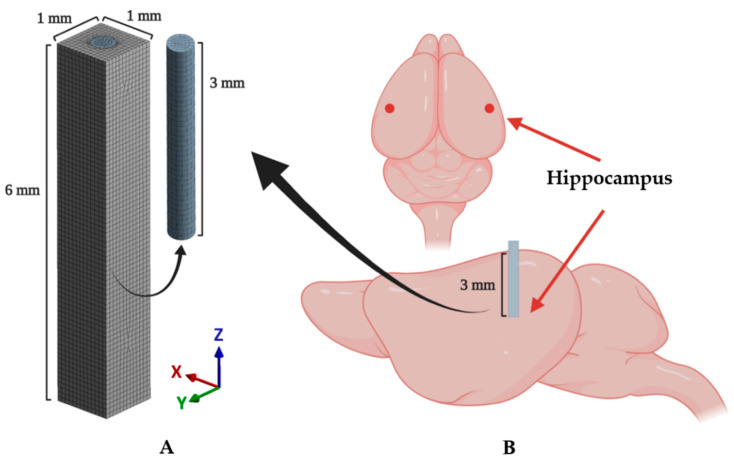
Schematics of the implantation and implanted model. (**A**) A 3D finite element (FE) model including mesh configuration. (**B**) An illustration of the implanted mouse brain. The target region (hippocampal region: mediolateral (ML): −3, anteroposterior (AP): −2.8, dorsoventral (DV): 3 (mm)) is marked as the red point in the above figure.

**Figure 4 micromachines-12-00394-f004:**
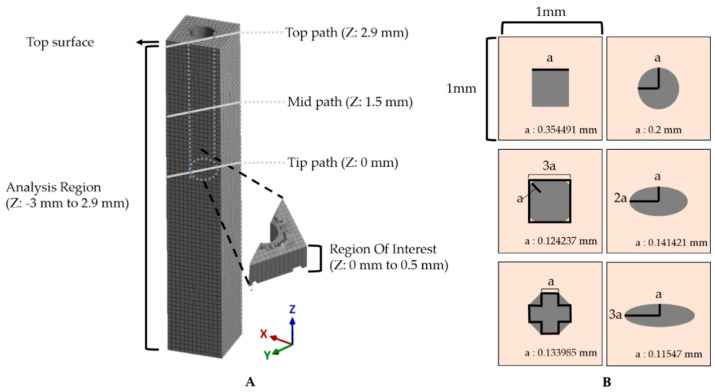
Method for assessment of simulation results. (**A**) The maximum and averaged equivalent strain versus time were investigated in the ROI (500 μm from the tip). Gray solid lines represent three paths: top (Z: 2.9 mm), mid (Z: 1.5 mm), and tip (Z: 0 mm). (**B**) Various cross-sections of the probe. The outer lightly colored square is a cross-section of the brain tissue model, and the inner shapes depict various designs; ellipses with an aspect ratio of 1:1/1:2/1:3, square, rounded square, and octagon. These are all designed to have an area of S = 0.04π μm^2^ as informed with the value of “a”.

**Figure 5 micromachines-12-00394-f005:**
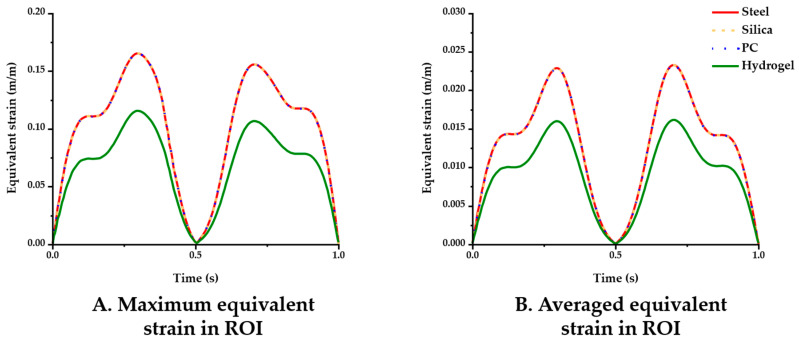
Equivalent strain variation versus time with varying probe materials: steel, silica, PC, and hydrogel (coefficient of friction (COF): 0.3). (**A**) Maximum equivalent strain; (**B**) averaged equivalent strain in the ROI.

**Figure 6 micromachines-12-00394-f006:**
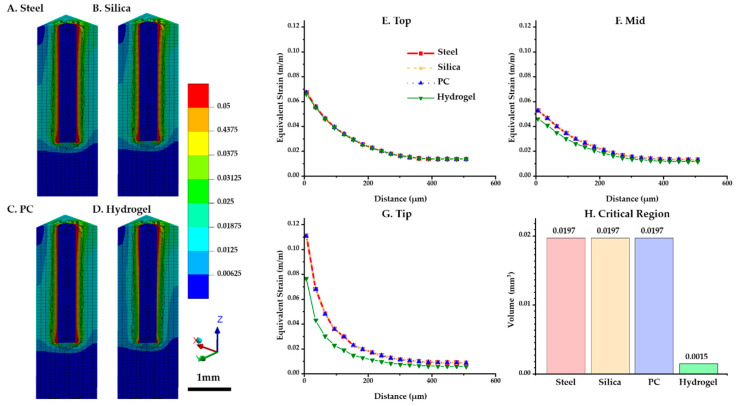
Maximum equivalent strain variation versus position with varying probe materials: steel, silica, PC, and hydrogel (COF: 0.3). (**A**–**D**) Strain distribution of the tissue and implant (steel, silica, PC, and hydrogel). (**E**–**G**) Equivalent strain profile in the brain tissue interfaced with each implant regarding three paths: top (at z = 2.9 mm), mid (at z = 1.5 mm), and tip (at z = 0 mm). (**H**) Volume of the critical region depending on materials.

**Figure 7 micromachines-12-00394-f007:**
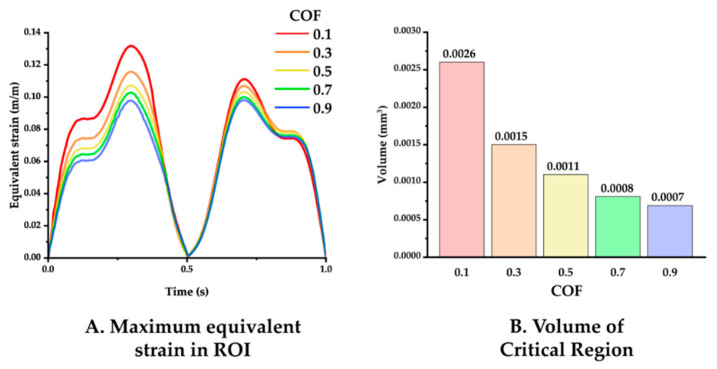
(**A**) Maximum equivalent strain variation versus time with varying COF: 0.1, 0.3, 0.5, 0.7, and 0.9 (probe material: hydrogel). (**B**) Volume of the critical region, depending on the COF.

**Figure 8 micromachines-12-00394-f008:**
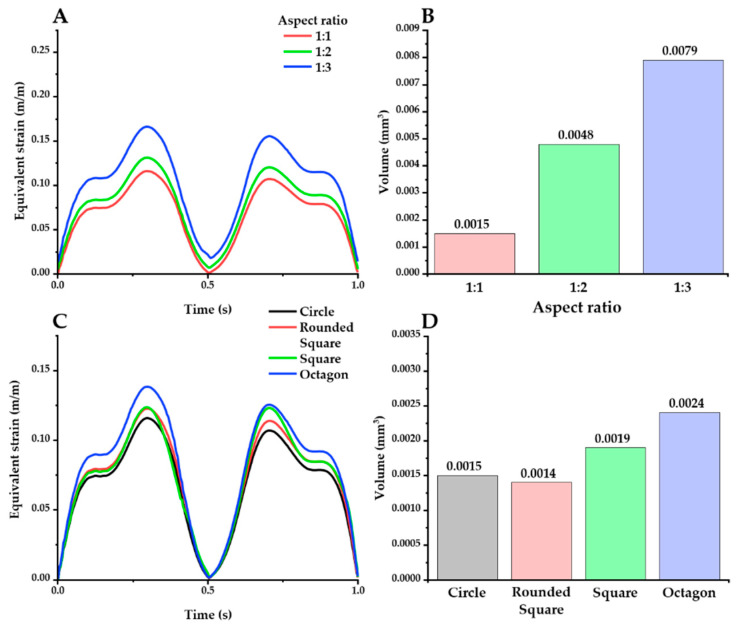
Equivalent strain distribution with varying aspect ratios (**A**,**B**) and shapes (**C**,**D**) (probe material: hydrogel). There are 2 kinds of plots for each factor: maximum equivalent strain versus time (**A**,**C**) and the volume of the critical region under different cross-section conditions (**B**,**D**).

**Figure 9 micromachines-12-00394-f009:**
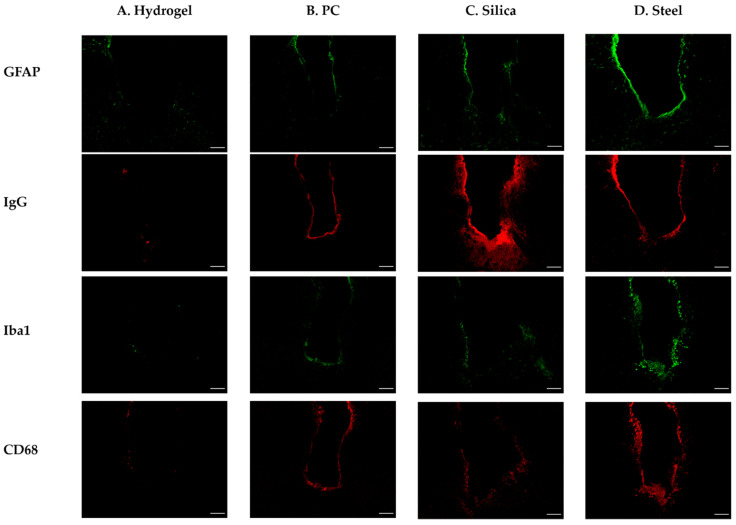
Representative confocal images of the immunohistochemistry with GFAP, IgG, Iba1, and CD68 after 4 weeks of different probe implantations. (**A**) Hydrogel; (**B**) PC; (**C**) silica; (**D**) steel. The white scale bar indicates 100 μm.

**Figure 10 micromachines-12-00394-f010:**
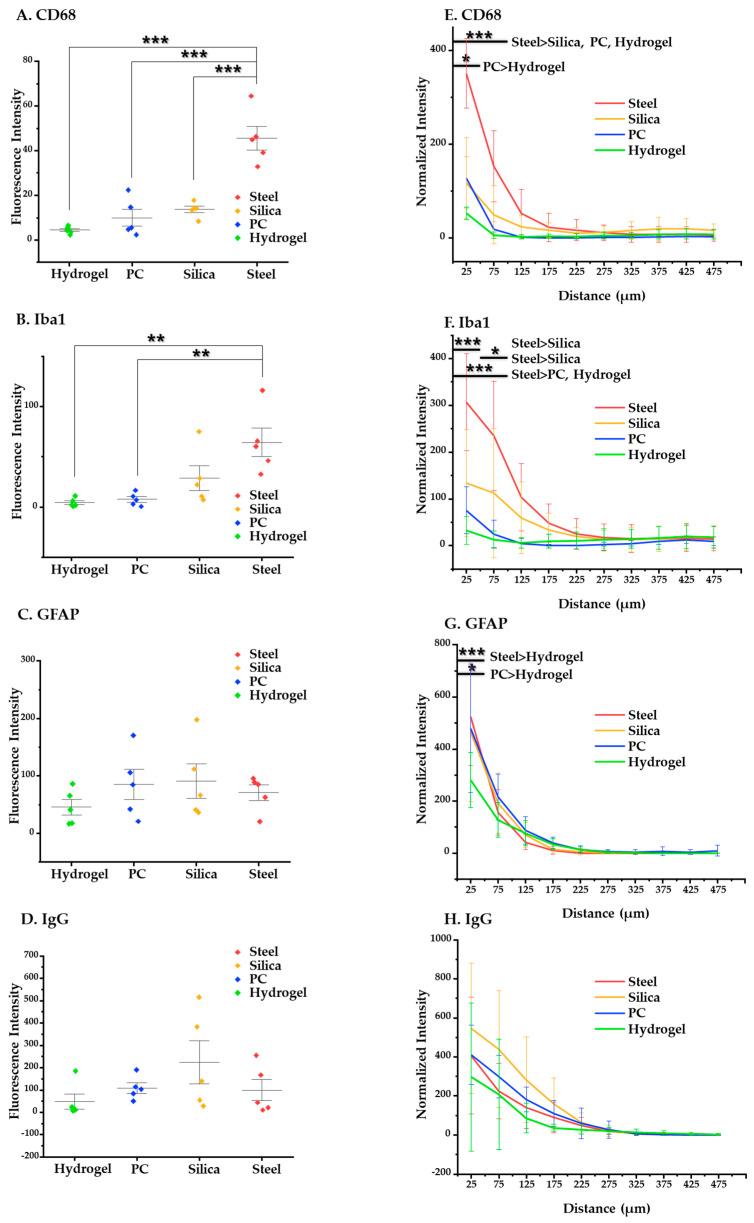
Statistical results of the chronic (4 weeks) immune response. (**A**–**D**) Results of the post-hoc Tukey’s test on one-way ANOVA with the mean values of intensity of each immune marker: (**A**) CD68; (**B**) Iba1; (**C**) GFAP; (**D**) IgG. (**E**–**H**) Results of post-hoc Tukey’s test on the two-way ANOVA based on distance, showing intensity profiles of each immune marker: (**E**) CD68, (**F**) Iba1, (**G**) GFAP, and (**H**) IgG. The error bars represent the standard deviation. Significance: *: *p* < 0.05, **: *p* < 0.01, and ***: *p* < 0.001.

**Figure 11 micromachines-12-00394-f011:**
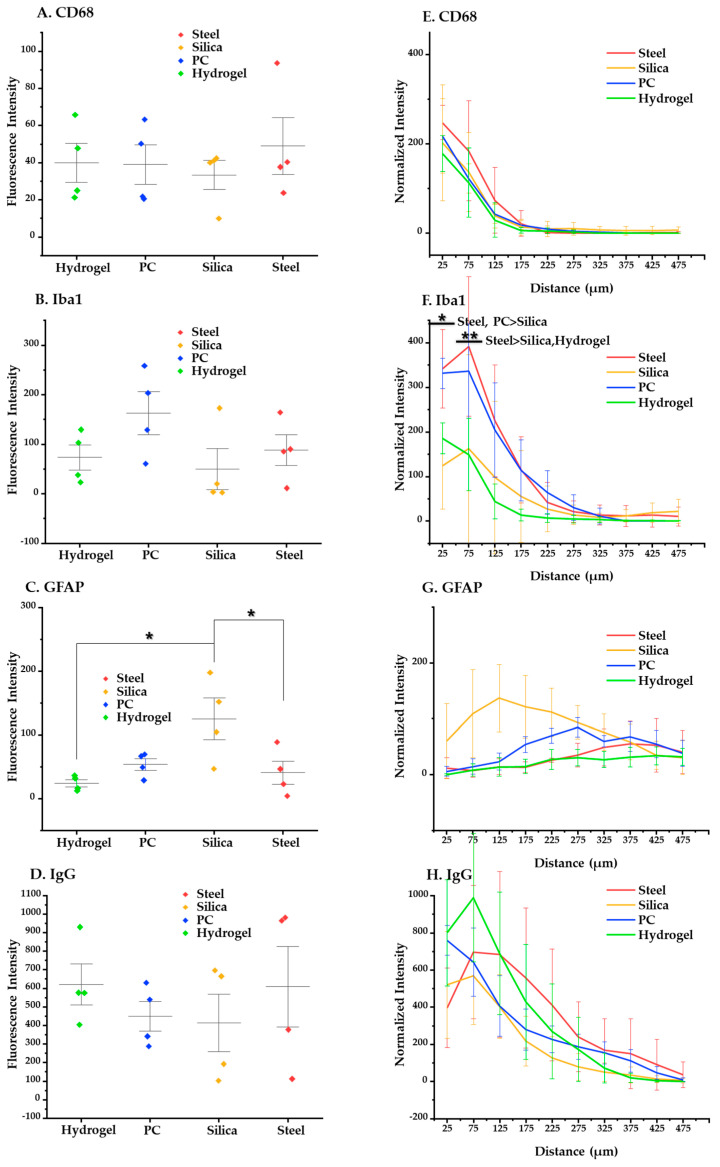
Statistical results about the acute (3 days) immune response. (**A**–**D**) Results of post-hoc Tukey’s test on one-way ANOVA with the mean values of intensity of each immune marker. (**A**) CD68, (**B**) Iba1, (**C**) GFAP, and (**D**) IgG. (**E**–**H**) Results of Tukey’s post-hoc test on two-way ANOVA based on distance, showing intensity profiles of each immune marker: (**E**) CD68; (**F**) Iba1; (**G**) GFAP; (**H**) IgG. The error bars represent the standard deviation. Significance: *: *p* < 0.05, **: *p* < 0.01.

**Figure 12 micromachines-12-00394-f012:**
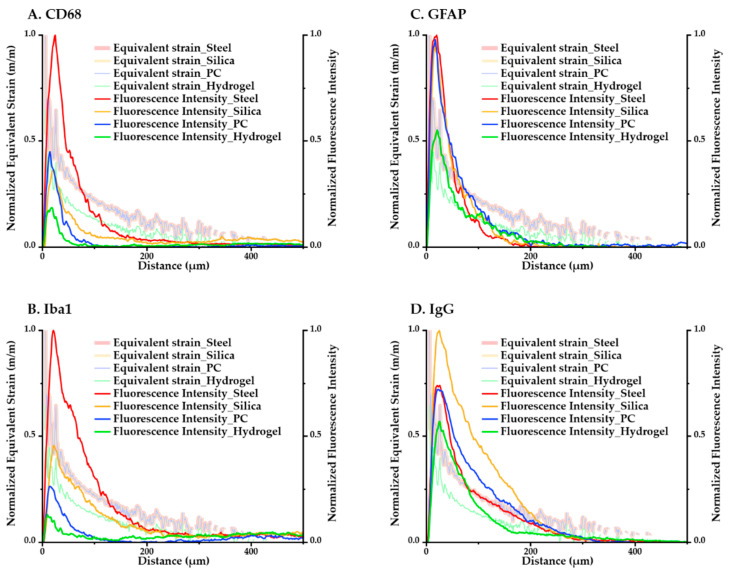
Overlay of min–max normalized equivalent strain plots and fluorescence intensity plots of each immune marker for all four kinds of materials. (**A**) CD68; (**B**) Iba1; (**C**) GFAP; (**D**) IgG.

**Figure 13 micromachines-12-00394-f013:**
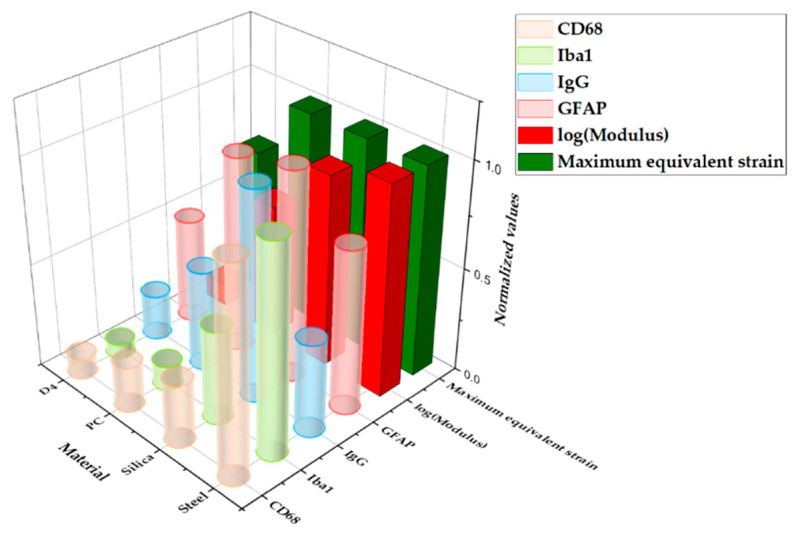
Three-dimensional (3D) plot summarizing the main results from the computational and histological analyses varying the base materials: steel, silica, PC, and hydrogel. All values were normalized with the maximum value in the group.

**Table 1 micromachines-12-00394-t001:** The detailed specification of the finite element model.

Component	Element Type	Number of Elements	Total Number of Elements	Total Number of Nodes	DOFs (Degrees of Freedom)
Brain tissue	SOLID187 ^1^	11,299	71,930	240,125	720,375
	SOLID186 ^2^	60,631			
Probe	SOLID187	103	1985	7881	23,643
	SOLID186	1882			

^1^ 10-node tetrahedral element in ANSYS. ^2^ 20-node hexahedral element in ANSYS.

**Table 2 micromachines-12-00394-t002:** Mechanical properties of the probe materials used in the finite element analysis.

Material	Young’s Modulus (Pa)	Poisson’s Ratio	Density (kg·m^−3^)
Steel ^1^	$\;1.93\times 10^{11}$	0.25	7990
Silica ^1^	$6.63\times 10^{10}$	0.15	2170
PC ^1^	$2.30\times 10^{9}$	0.37	1200
Hydrogel ^2^	$3.76\times 10^{6}$	0.46	1080

^1^ Material information provided by the supplier. ^2^ Young’s modulus is provided by the supplier.

**Table 3 micromachines-12-00394-t003:** The combination of primary and secondary antibodies used in this study.

Target	Primary Antibodies	Secondary Antibodies
Astrocytes	Anti-GFAP (1:1000; ab53554)	Donkey antigoat labeled with Alexa Fluor 488 (1:500; A11055)
Activated microglia/macrophages	Anti-CD68 (1:250; ab125212)	Donkey antigoat labeled with Alexa Fluor 488 (1:1000; A11055)
BBB breach	Donkey antimouse IgG conjugated to Alexa Fluor 568 (1:1000; A10037)	-
All microglia/macrophages	Anti-Iba1 (1:500; ab107159)	Donkey antirabbit labeled with Alexa Fluor 594 (1:1000; A21207)

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
