# Peer review of "Computational and Histological Analyses for Investigating Mechanical Interaction of Thermally Drawn Fiber Implants with Brain Tissue"

_micromachines, 2021, doi:10.3390/mi12040394_

Round 1
Reviewer 1 Report
I find this manuscript clearly written while using scientific terminologies. It reintroduces thermally drawn fiber-based neural implants to readers and then it presents the most biocompatible fiber by comparing a few fibers by their materials and design factors. I also found the title clear and straight. However, there are a few minor grammar errors that need to be addressed. I have pointed to some of the errors here.
30: the effects of the coefficient of friction
34: the implants with higher COF and/or circular cross-sections
77: inner shape depicts the cross-section of the neural probe.
54: these factors of a/the macroscopic preform
89: the first solvent casting
119: which show a/the superior solution
188: to be relevant to the chronic immune response
207: maximum characteristic length
241: perpendicular to the surface of the brain
253: the anti-mouse IgG antibody was dealt with as the secondary antibody.
258: transcardially perfused
265: Following of the primary incubation
275: MATLAB algorithm (Mathworks) was used
278: from the bottom of the implant site which was manually defined and was stacked
287: regions more than 500μm away from the probe surface. The ROIs were then binned into
290: a wide region that includes the most of the implant.
380: intensity profiles by the distance of each immune marker
462: cross-section geometry with a lower aspect ratio
Reviewer 2 Report
In general, the manuscript is well written, and presents the effect of soft material with lower young`s modulus to improve the mechanical mismatch between the tissue and the neural probes and foreign body response. For this, four different materials were chosen as a comparison study both with simulation and histological analyses.
Here are some comments for the manuscripts:
1) Why 400 ± 30 mm of the diameter has been chosen considering the diameter of a commonly used commercial Tungsten microwire is about 75 mm? How does this 30 mm fabrication variation affect to the histological results?
2) What are the young`s modulus of each material? You may be able to plot the histological results versus their young`s modulus?
3) All of the dummy implant pieces were packaged with PDMS, and how did you control the thickness of the PDMS layer? If you dip-coated the devices, its layer thickness can be varied sample by sample.
4) Also, the PDMS is a hydrophilic material. Was there any hermetic failure issue? Have you checked the extracted devices and seen if there was any corrosion for the metal piece?
5) If all the material were coated with the same PDMS, the interface between the tissue and the dummy implant would have been the same PDMS for all the experimental groups. How is the mechanical mismatch different of each experiment group different?
6) Please refer to the below bioRxiv paper:
- C. –Lombarte et al., bioRxiv, 2019
7) Have you done an analysis on the inflammation or fibrosis around the implant site versus time? Maybe with a non-invasive FMT imaging?
8) What are the main cause of the difference between different materials? Are they from acute scar formation and inflammation? Or are they related to chronic inflammation as well?
9) Where is the Sham group?
10) There is a study on geometrical design versus strain with simulations, but where is the related histological study result?
11) Visual presentation of some graphs can be further improved. Fonts or line widths of some graphs are quite small but some are not.
Reviewer 3 Report
The authors present a comparison of steel, silica, polycarbonate, hydrogel, thermally drawn fiber microelectrode mimics. Their outcomes include a FEM analysis of strains and effective volumes (regions strained above a critical 0.05 level). They also assess post mortem histology of the brain neuroinflammatory response after implantation of each probe in mice brains.
They conclude from modeling that the critically affected volume is not signficantly changed, except for the most flexible hydrogel probe. The authors also, reasonably conclude that coefficient of friction is inversely correlated with strain as is aspect ratio of the probe cross-section.
From their histological studies, they see that steel has the highest response (particularly with CD68) while hydrogel has the lowest across stains for BBB permeability, macrophages and microglia, and activated astrocytes.
Comments:
1) Cleaning and sterilization procedures should be discussed as these have been documented to substantially influence neuroinflammatory response. Ravikumar 2014
2) Implanation of the hydrogel electrode should be discussed, as it has been our experience that softer electrodes commonly require shuttles or other techniques to facilitate insertion.
3) Details regarding the hydrogel should be discussed generally -- what is polymer structure, what is swelling ratio, how hydrolytically stable is it?
4) Significant digits/decimals should be reviewed, e.g., 0.13176, 0.11565, 0.10701, 0.10254, 0.09792 -- are the extra digits meaningful w.r.t. the critical value of 0.05 selected? 0.00001 is 0.02% of the critical value...
5) With the minimal inflammatory response observed with the hydrogel, the authors should discuss how regions of electrode explant were identified. In the absence of staining how was the hole ROI defined?
Minor Comments
6) Colors should match left to right in figures 10, 11.
7) Minor English/grammar check required.
Round 2
Reviewer 2 Report
The authors have addressed most of my comments, and reflected those on the manuscripts to some extents.
As one last comment, I would like to suggest the authors to extend their discussion on the acute or chronic tissue response depending on four different types of materials instead of stopping at the assumption as written in page 14-15.
"For immunostaining at 3 days post-implantation (Figure 11), there was no strong evidence for the correlation between the immunoreactivity and elastic modulus of base-materials. While it is found that there are slightly higher microglia and macrophage recruitment near the stiffer devices, the result was not significant. We assumed that this is due to the wound healing process being more dominant right after the initial tissue penetration."
This may be an obvious result with an intuition, but suggestion on a better experiment setup can be added for a clear verification on this assumption. I guess the 3 days post implantation and 4 weeks post implantation groups are different, so...
If the authors think this does not need to be addressed further, you can ignore this comment.
